# Big Data, Leverage Scores, and Minimum Volume Covering Ellipsoids: Bridging Theory With Practice

## Abstract

The Minimum Volume Covering Ellipsoid (MVCE) problem, characterized by $n$ observations in $d$ dimensions where $n \gg d$, can be computationally very expensive in the big data regime. We apply methods from randomized numerical linear algebra to develop a data-driven leverage score sampling algorithm for solving MVCE, and establish theoretical error bounds and a convergence guarantee. Assuming the leverage scores follow a power law decay, we show that the computational complexity of computing the approximation for MVCE is reduced from $\mathcal{O}(nd^2)$ to $\mathcal{O}(nd \log n + \text{poly}(d))$, which is a significant improvement in big data problems. Numerical experiments on large-scale synthetic data, as well as real-world data, demonstrate the efficacy of our new algorithm, showing that it substantially reduces computation time while yielding near-optimal solutions.

## 1 Introduction

The Minimum Volume Covering Ellipsoid (MVCE) problem arises in many applied and theoretical areas. Statistical applications include outlier detection (Titterington, 1978), clustering (Rosen, 1965), and the closely related D-optimal design problem (Silvey, 1980). In fact, the MVCE problem and the D-optimal design problem are dual to one another (Sibson, 1972; Titterington, 1975). Containing ellipsoids are used in parameter identification and control theory to describe uncertainty sets for parameters and state vectors (Schweppe, 1968; Chernousko, 2005). Minimum volume covering ellipsoids are also used in computational geometry and computer graphics (Eberly, 2006), in particular, for collision detection (Chen et al., 2016).

Many algorithms for computing MVCEs and D-optimal designs have been studied in the literature. These include Frank-Wolfe type algorithms (Frank et al., 1956; Wolfe, 1970; Atwood, 1973; Khachiyan, 1996; Kumar & Yildirim, 2005), interior point algorithms (Khachiyan & Todd, 1993; Nesterov & Nemirovskii, 1994), the Dual Reduced Newton algorithm (Sun & Freund, 2004), the Cocktail algorithm (Yu, 2011), the Randomized Exchange algorithm (Harman et al., 2020), and the Fixed Point algorithm (Cohen et al., 2019; Song et al., 2022; Woodruff & Yasuda, 2023; 2025). However, when these algorithms are applied to very large datasets, they may be computationally inefficient, and may exceed storage limitations (Harman et al., 2020). One solution is to combine such solution algorithms with active set or batching strategies (Sun & Freund, 2004; Källberg & Andrén, 2019; Kudela, 2019; Rosa & Harman, 2022). The main idea of this approach is to iteratively apply the solution algorithm to a smaller subset of points until convergence to the solution. The motivation for this approach is that an ellipsoid is only determined by at most $d(d+3)/2$ points on its boundary (John, 2014).

Instead of applying an active set strategy, we apply deterministic sampling to reduce the number of points considered by the algorithm. This deterministic sampling method selects points corresponding to the highest statistical leverage scores. The resulting compressed dataset approximately maintains many of the qualities of the original dataset, provided that the number of samples is sufficiently large (Papailiopoulos et al., 2014; McCurdy et al., 2019). However, there is no immediate guarantee of the quality of solution to the MVCE problem for this compressed dataset.

In this paper, we present a new simplified proof of the subspace embedding theorem from McCurdy et al. (2019). Using this theorem, we provide the first theoretical guarantees on the quality of initial

and final solution for the MVCE problem when using deterministic leverage score sampling. Further, we show that these guarantees still hold (with high probability) when approximate leverage scores are used. Assuming a power law decay on the leverage scores, we show that our method improves the theoretical computation time required to approximate the MVCE. We also demonstrate the efficiency of our approach on synthetic and real world datasets.

**Notation.**  Throughout this paper, vectors and matrices are denoted by bold lowercase and bold uppercase letters, respectively (e.g. $\boldsymbol{a}$ and $\boldsymbol{A}$). The $i$th entry of $\boldsymbol{a}$ is denoted $a_i$, and the $(i,j)$th entry of $\boldsymbol{A}$ is denoted $\boldsymbol{A}_{ij}$. Let $\boldsymbol{A}$ and $\boldsymbol{B}$ be symmetric positive definite matrices, then $\boldsymbol{A} \succeq \boldsymbol{B}$ if $\boldsymbol{A} - \boldsymbol{B}$ is symmetric positive semidefinite; and $\boldsymbol{A} \succ \boldsymbol{B}$ if $\boldsymbol{A} - \boldsymbol{B}$ is symmetric positive definite. The determinant of a matrix $\boldsymbol{A}$ is denoted $\det(\boldsymbol{A})$. Unless otherwise specified, we use $\boldsymbol{U} = \text{diag}(\boldsymbol{u})$, where $\boldsymbol{U}$ is a matrix with the vector $\boldsymbol{u}$ in the main diagonal. We use regular lowercase to denote scalar constants (e.g. $c$). Finally, $\boldsymbol{e}$ denotes a vector of ones, and $\boldsymbol{e}_i$ denotes the vector with one at position $i$, and zero otherwise.

## 2 Minimum Volume Covering Ellipsoid Problem

Let $\mathcal{X} = \{\boldsymbol{x}_1, \ldots, \boldsymbol{x}_n\}$ be a set of data points in $\mathbb{R}^d$. Then the minimum volume covering ellipsoid is the ellipsoid that covers $\mathcal{X}$, which attains the minimum volume of all covering ellipsoids of $\mathcal{X}$. We assume throughout that there exists a non-degenerate minimum volume covering ellipsoid.

We define the ellipsoid $\mathcal{E}(\boldsymbol{Q}, \boldsymbol{x}_c)$ as

$$\mathcal{E}(\boldsymbol{Q}, \boldsymbol{x}_c) := \{\boldsymbol{x} \in \mathbb{R}^d \,:\, (\boldsymbol{x} - \boldsymbol{x}_c)^\mathsf{T} \boldsymbol{Q} (\boldsymbol{x} - \boldsymbol{x}_c) \le d\},$$

where $\boldsymbol{x}_c$ is the center of the ellipsoid, and $\boldsymbol{Q}$ is a $d$-dimensional symmetric positive definite matrix. Its volume is given by

$$\text{vol}(\mathcal{E}(\boldsymbol{Q}, \boldsymbol{x}_c)) = d^{d/2} \Omega_d \det(\boldsymbol{Q})^{-1/2},$$

where $\Omega_d$ is the volume of the unit ball in $\mathbb{R}^d$ (e.g., see Todd (2016a)).

We can now write the mathematical formulation of the Minimum Volume Covering Ellipsoid problem. Suppose we have a finite set of points $\mathcal{X} = \{\boldsymbol{x}_1, \ldots, \boldsymbol{x}_n\} \subset \mathbb{R}^d$. Then its minimum volume covering ellipsoid can be found by solving

$$\underset{\boldsymbol{Q} \succ \boldsymbol{0},\, \boldsymbol{x}_c \in \mathbb{R}^d}{\text{minimize}} \quad -\log\det(\boldsymbol{Q}) \tag{$\text{P}_0$}$$
$$\text{subject to} \quad (\boldsymbol{x}_i - \boldsymbol{x}_c)^\mathsf{T} \boldsymbol{Q} (\boldsymbol{x}_i - \boldsymbol{x}_c) \le d, \quad i = 1, \ldots, n.$$

Although the objective function is convex, ($\text{P}_0$) itself is not convex (e.g., see Todd (2016a)).

We therefore reformulate ($\text{P}_0$) so that it is convex. At the cost of working in $\mathbb{R}^{d+1}$, we can calculate the centered minimum volume covering ellipsoid, and recover the solution to ($\text{P}_0$) (Titterington, 1975). Therefore, we can set $\boldsymbol{x}_c = \boldsymbol{0}$, and obtain

$$\underset{\boldsymbol{Q} \succ \boldsymbol{0}}{\text{minimize}} \quad f(\boldsymbol{Q}) := -\log\det(\boldsymbol{Q}) \tag{P}$$
$$\text{subject to} \quad \boldsymbol{x}_i^\mathsf{T} \boldsymbol{Q} \boldsymbol{x}_i \le d, \quad i = 1, \ldots, n.$$

We let $\boldsymbol{Q}^*$ denote an optimal solution to (P). Then $\text{MVCE}(\mathcal{X}) := \mathcal{E}(\boldsymbol{Q}^*, \boldsymbol{0})$ is the minimum volume covering ellipsoid. We will refer to (P) as the MVCE problem.

The dual problem to (P) is the D-optimal design problem. This problem is concave, and can be formulated as

$$\underset{\boldsymbol{u} \in \mathbb{R}^n}{\text{maximize}} \quad g(\boldsymbol{u}) := \log\det\left(\sum_{i=1}^{n} u_i \boldsymbol{x}_i \boldsymbol{x}_i^\mathsf{T}\right) \tag{D}$$
$$\text{subject to} \quad \sum_{i=1}^{n} u_i = 1, \quad \boldsymbol{u} \ge \boldsymbol{0},$$

where $\boldsymbol{u}$ is called the design vector. We note that for every design vector $\boldsymbol{u}$, we can find its associated shape matrix

$$\boldsymbol{Q}\left(\boldsymbol{u}\right) := \left(\sum_{i=1}^{n} u_i \boldsymbol{x}_i \boldsymbol{x}_i^{\mathsf{T}}\right)^{-1},$$

provided that the inverse exists. Hence if we have an optimal solution $\boldsymbol{u}^*$ to (D), then the optimal solution to (P) is $\boldsymbol{Q}\left(\boldsymbol{u}^*\right)$.

We note that (D) (and (P)) cannot usually be solved exactly, so we will focus on deriving approximate solutions. To ensure the chosen algorithm terminates with a guaranteed quality of solution, we will define some approximate optimality conditions. A feasible $\boldsymbol{u}$ for (D) is called $\delta$-primal feasible if $\boldsymbol{Q}\left(\boldsymbol{u}\right)$ satisfies

$$\boldsymbol{x}_i^{\mathsf{T}} \boldsymbol{Q}\left(\boldsymbol{u}\right) \boldsymbol{x}_i \leq (1 + \delta) \, d,$$

for all $i = 1, \ldots, n$. If $\boldsymbol{Q}\left(\boldsymbol{u}\right)$ additionally satisfies

$$\boldsymbol{x}_i^{\mathsf{T}} \boldsymbol{Q}\left(\boldsymbol{u}\right) \boldsymbol{x}_i \geq (1 - \delta) \, d \text{ if } u_i > 0,$$

for all $i = 1, \ldots, n$, then we say that $\boldsymbol{u}$ is $\delta$-approximately optimal. These optimality conditions ensure the optimality gap is small.

**Proposition 1** (Todd (2016a), Proposition 2.9). *If we have a $\delta$-primal feasible (or $\delta$-approximately optimal) solution $\boldsymbol{u}$, then $\boldsymbol{u}$ and $(1 + \delta)^{-1} \boldsymbol{Q}\left(\boldsymbol{u}\right)$ are both within $d \log \left(1 + \delta\right)$ of being optimal in (D) and (P), respectively.*

## 3 SAMPLING USING LEVERAGE SCORES

The concept of statistical leverage scores has long been used in statistical regression diagnostics to identify outliers (Rousseeuw & Hubert, 2011). Given a data matrix $\boldsymbol{X} \in \mathbb{R}^{n \times d}$ with $n > d$, consider any orthogonal matrix $\boldsymbol{A}$ such that $\mathrm{Range}(\boldsymbol{A}) = \mathrm{Range}(\boldsymbol{X})$. The $i$th leverage score corresponding to the $i$th row of $\boldsymbol{X}$ is defined as

$$\ell_i \left(\boldsymbol{X}\right) := \|\boldsymbol{A}(i,:)\|_2^2.$$

It can be easily shown that this is well defined in that the leverage score does not depend on the particular choice of the basis matrix $\boldsymbol{A}$. Furthermore, the $i$th leverage score is the $i$th diagonal entry of the Hat matrix, that is,

$$\ell_i \left(\boldsymbol{X}\right) = \boldsymbol{e}_i^{\mathsf{T}} \boldsymbol{H} \boldsymbol{e}_i, \qquad i = 1, \ldots, n,$$

where

$$\boldsymbol{H} := \boldsymbol{X}(\boldsymbol{X}^{\mathsf{T}} \boldsymbol{X})^{-1} \boldsymbol{X}^{\mathsf{T}}.$$

The Hat matrix is symmetric and idempotent. We can use these properties to easily show that

$$0 \leq \ell_i \left(\boldsymbol{X}\right) \leq 1,$$

for all $i$, and

$$\sum_{i=1}^{n} \ell_i \left(\boldsymbol{X}\right) = \mathrm{rank}\left(\boldsymbol{X}\right).$$

Since we only consider full rank matrices, the sum of the leverage scores is $d$.

### 3.1 DETERMINISTIC SAMPLING

When sampling deterministically, we sample the $s$ rows from $\boldsymbol{X}$ with highest leverage scores. Without loss of generality, assume that the leverage scores are ordered $\ell_1 \left(\boldsymbol{X}\right) \geq \cdots \geq \ell_n \left(\boldsymbol{X}\right)$. We summarize the sampling procedure in Algorithm 1.

Using this algorithm, we obtain the following subspace embedding result.

---

**Algorithm 1** Deterministic Leverage Score Sampling with Threshold (Papailiopoulos et al., 2014)

---

**Require:** $\boldsymbol{X} = [\boldsymbol{x}_1, \dots, \boldsymbol{x}_n]^\mathsf{T} \in \mathbb{R}^{n \times d}, \varepsilon \in (0, 1)$
 1: Calculate leverage scores for each row in $\boldsymbol{X}$. Assume $\ell_1(\boldsymbol{X}) \geq \cdots \geq \ell_n(\boldsymbol{X})$.
 2: Let $s = \arg\min_j \left( \sum_{i=1}^{j} \ell_i(\boldsymbol{X}) > d - \varepsilon \right)$.
 3: Let $\boldsymbol{R} = \boldsymbol{0} \in \mathbb{R}^{s \times n}$.
 4: **for** $i = 1 : s$ **do**
 5:     Set row $i$ of $\boldsymbol{R}$ equal to $\boldsymbol{e}_i$.
 6: **end for**
**Ensure:** $\boldsymbol{R}, s$

---

**Theorem 2** (McCurdy et al. (2019), Theorem 1). *Let $\varepsilon \in (0, 1)$. Use Algorithm 1 to construct the sampling matrix $\boldsymbol{R}$. Then*

$$(1 - \varepsilon)\boldsymbol{X}^\mathsf{T}\boldsymbol{X} \prec (\boldsymbol{R}\boldsymbol{X})^\mathsf{T}\boldsymbol{R}\boldsymbol{X} \preceq \boldsymbol{X}^\mathsf{T}\boldsymbol{X}.$$

We now present our new simplified proof of the lower bound.

*Proof.* We use the fact that for any $i$,

$$\boldsymbol{x}_i \boldsymbol{x}_i^\mathsf{T} \preceq \ell_i(\boldsymbol{X})\, \boldsymbol{X}^\mathsf{T}\boldsymbol{X},$$

see, for example, the proof of Lemma 4 in Cohen et al. (2015). Hence

$$\boldsymbol{X}^\mathsf{T}\boldsymbol{X} - (\boldsymbol{R}\boldsymbol{X})^\mathsf{T}\boldsymbol{R}\boldsymbol{X} = \sum_{i=s+1}^{n} \boldsymbol{x}_i \boldsymbol{x}_i^\mathsf{T} \preceq \sum_{i=s+1}^{n} \ell_i(\boldsymbol{X})\, \boldsymbol{X}^\mathsf{T}\boldsymbol{X} = \left( \sum_{i=s+1}^{n} \ell_i(\boldsymbol{X}) \right) \boldsymbol{X}^\mathsf{T}\boldsymbol{X} \prec \varepsilon \boldsymbol{X}^\mathsf{T}\boldsymbol{X},$$

since $\sum_{i=s+1}^{n} \ell_i(\boldsymbol{X}) < \varepsilon$, by construction of Algorithm 1. Rearranging, we obtain our lower bound. $\qquad\square$

Algorithm 1 has time complexity $\mathcal{O}\left(nd^2\right)$, due to the cost of calculating the leverage scores exactly. This can be improved by using approximate leverage score algorithms. For example, the algorithm of Drineas et al. (2012) uses a fast sampled randomized Hadamard transform (SRHT), and has time complexity $\mathcal{O}\left(nd \log n\right)$.[1] Moreover, we can use these approximate leverage score algorithms to prove an analogous result to Theorem 2 (see Appendix A).

We note that for many datasets, the number of points selected after applying Algorithm 1 may still be large. In this case, we can apply Algorithm 1 multiple times, to obtain a similar subspace embedding result (see Appendix B).

## 4  OUR APPROACH

We are interested in calculating MVCE($\mathcal{X}$), where $\mathcal{X} = \{\boldsymbol{x}_1, \dots, \boldsymbol{x}_n\} \subset \mathbb{R}^d$, $n \gg d$. For very large $n$ (and $d$), this can be very computationally expensive. Instead of calculating MVCE($\mathcal{X}$) directly, we will calculate the MVCE on a much smaller subset of $\mathcal{X}$. We select our subset by using leverage score based sampling, as introduced in Section 3. Let this subset be $\mathcal{X}_s$, where $\mathcal{X}_s$ contains up to $s$ points sampled from $\mathcal{X}$. For ease of calculation, let the points from $\mathcal{X}$ be stored in the rows of the data matrix $\boldsymbol{X} \in \mathbb{R}^{n \times d}$. Then we sample $s$ rows from $\boldsymbol{X}$, which we store in the rows of $\boldsymbol{X}_s \in \mathbb{R}^{s \times d}$. We then use $\boldsymbol{X}_s$ instead of $\boldsymbol{X}$ in our solution algorithm. That is, our solution algorithm solves the concave problem

$$\underset{\boldsymbol{u} \in \mathbb{R}^s}{\text{maximize}} \quad g_s(\boldsymbol{u}) := \log \det \left( \boldsymbol{X}_s^\mathsf{T} \boldsymbol{U} \boldsymbol{X}_s \right) \tag{$\text{D}_s$}$$

$$\text{subject to} \quad \sum_{i=1}^{s} u_i = 1, \quad \boldsymbol{u} \geq \boldsymbol{0}.$$

---

[1] In a recent preprint, Eshragh et al. (2023) developed the Sequential Approximate Leverage-Score Algorithm (SALSA) with time complexity $\mathcal{O}\left(nd\right)$.

### 4.1 Computational Complexity

We compare the computational complexity of our algorithm to the current state of the art algorithm, the Wolfe-Atwood (WA) algorithm (Wolfe, 1970; Atwood, 1973), which computes a $\delta$-approximately optimal solution to (D). It uses the Kumar-Yildirim initialization (Kumar & Yildirim, 2005), which places equal weight on a small subset of points. These algorithms have time complexity $\mathcal{O}\left(nd^2\left(\log\log d + \delta^{-1}\right)\right)$ (Todd & Yıldırım, 2007) and $\mathcal{O}\left(nd^2\right)$ (Kumar & Yildirim, 2005), respectively. Together, this costs $\mathcal{O}\left(nd^2\left(\log\log d + \delta^{-1}\right)\right)$.

The computational complexity of our approach is as follows. We calculate approximate leverage scores in $\mathcal{O}\left(nd\log n\right)$ time, which, with high probability, satisfy $\hat{\ell}_i\left(\boldsymbol{X}\right) = \left(1 \pm \beta\right)\ell_i\left(\boldsymbol{X}\right)$, for some $\beta \in \left(0, 1/2\right]$ (Drineas et al., 2012). Then Algorithm 1 also runs in $\mathcal{O}\left(nd\log n\right)$ time. We then use the WA algorithm with Kumar-Yildirim initialization (Kumar & Yildirim, 2005) to find an optimal solution to (D$_s$). Since $\boldsymbol{X}_s \in \mathbb{R}^{s \times d}$, these algorithms have time complexity $\mathcal{O}\left(sd^2\log\log d\right)$ and $\mathcal{O}\left(sd^2\right)$, respectively. Thus, the total time complexity is $O\left(nd\log n + sd^2\left(\log\log d + \delta^{-1}\right)\right)$. Further, if the leverage scores exhibit a power law decay, then $s = \text{poly}\left(d\left(1 + \beta\right), \frac{1}{(1-\beta)\varepsilon}\right)$ (see Appendix A).

### 4.2 Comparison With the Work of Cohen Et. Al.

Recently, Cohen et al. (2019) developed the Fixed Point algorithm, which computes a $\delta$-primal feasible solution to (D). (Woodruff & Yasuda, 2023) extend this result; by sampling rows of $\boldsymbol{X}$ with probabilities proportionate to the weights calculated by the Fixed Point algorithm, they obtain a $\delta$-approximately optimal solution to (D).

The computational complexity of the Fixed Point algorithm is as follows. We examine Algorithm 2 from Cohen et al. (2019), which uses sketching techniques from randomized numerical linear algebra to speed up each iteration. Theorem C.7 from Cohen et al. (2019) states that Algorithm 2 takes at most $O\left(\delta^{-1}\log\left(\frac{n}{d}\right)\right)$ iterations to complete. In each iteration there are $O\left(\delta^{-1}\right)$ linear systems of the form $\boldsymbol{A}^\mathsf{T}\boldsymbol{W}\boldsymbol{A}\boldsymbol{x} = \boldsymbol{b}$ to be solved. Assuming $\boldsymbol{A}$ is dense, solving each of these linear systems costs $O(nd)$. This gives a total time complexity of $O\left(\delta^{-2}nd\log\left(\frac{n}{d}\right)\right)$ for dense matrices.

In big data regimes with $n \gg d$, it is reasonable to assume that $\delta < \frac{1}{d}$ is desirable. This assumption holds for many problem instances considered in the MVCE literature, since these problems typically have dimension $d \leq 200$, and are solved until tolerance $\delta = 10^{-6}$ (or stricter) is achieved (see, e.g. Sun & Freund (2004); Damla Ahipasaoglu et al. (2008); Yu (2011); Källberg & Andrén (2019); Kudela (2019)). With this assumption, Algorithm 2 has total time complexity $O\left(nd^3\log(nd)\right)$ and our algorithm has total time complexity $O\left(nd\log n + sd^3\right)$, where $s \leq n$. Thus, in the context of tall data matrices with dense input, our algorithm outperforms the Fixed Point algorithm theoretically. (This also holds numerically, see Appendix E.)

## 5 Initial Optimality Gap

We would like to know how well MVCE($\mathcal{X}_s$) approximates MVCE($\mathcal{X}$). We will first provide an upper bound for the initial optimality gap, for a particular choice of initial $\boldsymbol{u}$. This $\boldsymbol{u}$ must be feasible for (D$_s$), that is, we must have $\sum_{i=1}^s u_i = 1$, and $\boldsymbol{u} \geq \boldsymbol{0}$. We choose

$$\boldsymbol{u}_0 = \left[\tfrac{1}{s}, \ldots, \tfrac{1}{s}\right]^\mathsf{T} \in \mathbb{R}^s.$$

Then the initial optimality gap is given by

$$g^* - g_s(\boldsymbol{u}_0),$$

where $g^*$ is the optimal objective value when the full set $\mathcal{X}$ is considered.

To derive our bound, we will compare our initial solution with an initialization due to Khachiyan (1996). Khachiyan's initialization $\boldsymbol{u}_K$ puts equal weight on all $n$ points of $\mathcal{X}$, and guarantees that

$$g^* - g(\boldsymbol{u}_K) \leq d\log n,$$

as shown by Khachiyan (1996). Therefore, our bound can be found by exploiting the fact that

$$g^* - g_s(\boldsymbol{u}_0) = (g^* - g(\boldsymbol{u}_K)) + (g(\boldsymbol{u}_K) - g_s(\boldsymbol{u}_0)).$$

Upon simplification,

$$g(\boldsymbol{u}_K) = \log \det \left( \boldsymbol{X}^\mathsf{T} \boldsymbol{X} \right) - d \log n, \tag{1}$$

and, similarly,

$$g_s(\boldsymbol{u}_0) = \log \det \left( \boldsymbol{X}_s^\mathsf{T} \boldsymbol{X}_s \right) - d \log s. \tag{2}$$

Hence

$$g^* - g_s(\boldsymbol{u}_0) \leq \log \det \left( \boldsymbol{X}^\mathsf{T} \boldsymbol{X} \right) - \log \det \left( \boldsymbol{X}_s^\mathsf{T} \boldsymbol{X}_s \right) + d \log s. \tag{3}$$

This suggests the need to compare $\log \det \left( \boldsymbol{X}_s^\mathsf{T} \boldsymbol{X}_s \right)$ with $\log \det \left( \boldsymbol{X}^\mathsf{T} \boldsymbol{X} \right)$.

Suppose we construct $\boldsymbol{X}_s = \boldsymbol{R} \boldsymbol{X}$ as in Algorithm 1. Then Theorem 2 guarantees that

$$(1 - \varepsilon) \boldsymbol{X}^\mathsf{T} \boldsymbol{X} \prec \boldsymbol{X}_s^\mathsf{T} \boldsymbol{X}_s.$$

Now, for positive semidefinite matrices $\boldsymbol{A}$, $\boldsymbol{B}$, with $\boldsymbol{A} \preceq \boldsymbol{B}$, we have

$$\det \boldsymbol{A} \leq \det \boldsymbol{B}. \tag{4}$$

Since the logarithm is monotonic, it then follows that $\log \det \boldsymbol{A} \leq \log \det \boldsymbol{B}$. Hence

$$\log \det \left( \boldsymbol{X}^\mathsf{T} \boldsymbol{X} \right) < \log \det \left( \boldsymbol{X}_s^\mathsf{T} \boldsymbol{X}_s \right) - d \log(1 - \varepsilon). \tag{5}$$

Combining Equations (3) and (5), we obtain the bound presented in Theorem 3.

**Theorem 3.** *Select $\boldsymbol{X}_s = \boldsymbol{R} \boldsymbol{X}$ using Algorithm 1, and let our initial solution be given by $\boldsymbol{u}_0 = \frac{1}{s} \boldsymbol{e}$. Then*

$$g^* - g_s(\boldsymbol{u}_0) < d \log \left( \frac{s}{1 - \varepsilon} \right).$$

*Proof.* See Appendix C. □

## 6 FINAL OPTIMALITY GAP

We now provide an upper bound for the final optimality gap. The final optimality gap is given by

$$g^* - g_s^*,$$

where $g_s^*$ is the optimal objective value when only $\mathcal{X}_s$ is considered. Note that for any $\boldsymbol{u}$ feasible for $\mathrm{D}_s$, we have

$$g^* - g_s^* \leq g^* - g_s(\boldsymbol{u}),$$

since $g_s$ is concave.

Consider the feasible solution $\tilde{\boldsymbol{u}}_s$ for $\mathrm{D}_s$, given by

$$\tilde{\boldsymbol{u}}_s = \frac{1}{\boldsymbol{e}^\mathsf{T} \boldsymbol{u}_s} \boldsymbol{u}_s,$$

where $\boldsymbol{u}_s$ contains the first $s$ entries of $\boldsymbol{u}^*$. We would like a bound similar to the one in Theorem 2, with $\boldsymbol{X}$ replaced with a rescaled version $\boldsymbol{Y}$. More precisely, let

$$\boldsymbol{Y} := \sqrt{\boldsymbol{U}^*} \boldsymbol{X}.$$

To apply Theorem 2, we require

$$\sum_{i=s+1}^{n} \ell_i \left( \boldsymbol{Y} \right) < \varepsilon, \tag{6}$$

for some $\varepsilon \in (0, 1)$. To show this, we use the following result.

**Proposition 4.** *Use Algorithm 1 to construct $\boldsymbol{X}_s = \boldsymbol{R}\boldsymbol{X}$, with $\varepsilon \in (0,1)$. Suppose that an optimal solution $\boldsymbol{u}^*$ of (D) satisfies $u_i^* > 0$ for all $i = 1, \ldots, s$. Define $\boldsymbol{Y} := \sqrt{\boldsymbol{U}^*}\boldsymbol{X}$. Then*

$$\sum_{i=s+1}^{n} \ell_i\left(\boldsymbol{Y}\right) \leq \sum_{i=s+1}^{n} \ell_i\left(\boldsymbol{X}\right).$$

*Otherwise, let $\boldsymbol{u}$ be a $\delta$-feasible solution to (D). Define $\boldsymbol{Y} := \sqrt{\boldsymbol{U}}\boldsymbol{X}$. Then*

$$\sum_{i=s+1}^{n} \ell_i\left(\boldsymbol{Y}\right) \leq \sum_{i=s+1}^{n} \ell_i\left(\boldsymbol{X}\right).$$

*Proof.* See Appendix D. $\square$

Then (6) necessarily holds, by construction of Algorithm 1. Therefore, we may apply Theorem 2 with $\boldsymbol{Y}$ instead of $\boldsymbol{X}$, to obtain the bound

$$(1 - \varepsilon)\boldsymbol{X}^\mathsf{T}\boldsymbol{U}^*\boldsymbol{X} \prec \boldsymbol{X}_s^\mathsf{T}\boldsymbol{U}_s\boldsymbol{X}_s.$$

That is, the feasible solution $\tilde{\boldsymbol{u}}_s$ satisfies

$$(1 - \varepsilon)\boldsymbol{X}^\mathsf{T}\boldsymbol{U}^*\boldsymbol{X} \preceq \frac{1 - \varepsilon}{\boldsymbol{e}^\mathsf{T}\boldsymbol{u}_s}\boldsymbol{X}^\mathsf{T}\boldsymbol{U}^*\boldsymbol{X} \prec \frac{1}{\boldsymbol{e}^\mathsf{T}\boldsymbol{u}_s}\boldsymbol{X}_s^\mathsf{T}\boldsymbol{U}_s\boldsymbol{X}_s = \boldsymbol{X}_s^\mathsf{T}\tilde{\boldsymbol{U}}_s\boldsymbol{X}_s.$$

Hence

$$
\begin{aligned}
g^* - g_s^* &\leq g^* - g_s(\tilde{\boldsymbol{u}}_s) \\
&= \log\det\left(\boldsymbol{X}^\mathsf{T}\boldsymbol{U}^*\boldsymbol{X}\right) - \log\det\left(\boldsymbol{X}_s^\mathsf{T}\tilde{\boldsymbol{U}}_s\boldsymbol{X}_s\right) \\
&< \log\det\left(\boldsymbol{X}^\mathsf{T}\boldsymbol{U}^*\boldsymbol{X}\right) - \left(d\log(1 - \varepsilon) + \log\det\left(\boldsymbol{X}^\mathsf{T}\boldsymbol{U}^*\boldsymbol{X}\right)\right) \\
&= d\log\left(\frac{1}{1 - \varepsilon}\right).
\end{aligned}
$$

**Theorem 5.** *Use Algorithm 1 to construct $\boldsymbol{X}_s = \boldsymbol{R}\boldsymbol{X}$, with $\varepsilon \in (0,1)$. If there exists an optimal solution $\boldsymbol{u}^*$ of (D) satisfying $u_i^* > 0$ for all $i = 1, \ldots, s$, then*

$$g^* - g_s^* < d\log\left(\frac{1}{1 - \varepsilon}\right).$$

*Otherwise,*

$$g^* - g_s^* < d\log\left(\frac{1 + \delta}{1 - \varepsilon}\right),$$

*where the parameter $\delta > 0$ can be chosen to be arbitrarily small.*

*Proof.* See Appendix D. $\square$

## 7 NUMERICAL RESULTS

We generate three large datasets of size $n = 10^7$, $d = 100$. Without loss of generality, we assume the leverage scores are sorted in descending order, that is, $\ell_1\left(\boldsymbol{X}\right) \geq \cdots \geq \ell_n\left(\boldsymbol{X}\right)$. The first dataset is Rotated Cauchy (Todd, 2016a). The points are generated so that they have rotational symmetry, and the distances of the points from the origin are Cauchy. The leverage scores of these points quickly decay. The second dataset is Lognormal, which has a shallower leverage score decay. The third dataset is Gaussian, which has leverage scores that are close to uniform. We provide additional numerical results in Appendix E.

All computations are performed on a personal laptop with a 64 bit MacOS 13 operating system, and a 2.4 GHz Quad-Core Intel Core i5 processor with 8 GB of RAM. The algorithms are run using MATLAB (R2021a).

We use the WA algorithm (Wolfe, 1970; Atwood, 1973) to calculate the MVCEs of the two datasets, with $\delta = 10^{-9}$. We initialize using the Kumar-Yildirim initialization (Kumar & Yildirim, 2005). We use Todd's Matlab implementation of these algorithms (Todd, 2016b). Then, for $s$ varying from $0.1$ to $10\%$ of $n$, we sample from each dataset in three ways: deterministic leverage score sampling, uniform sampling, and randomized leverage score sampling (sampling with probabilities proportionate to the leverage scores (Drineas et al., 2008)). In these results, we use the exact leverage scores.

Let $\tilde{g}^*$ and $\tilde{g}^*_s$ be the optimal values obtained when using the WA algorithm on the full and sampled datasets respectively. In Figure 1a, we summarize the calculated optimality gaps $\tilde{g}^* - \tilde{g}^*_s$ for the Rotated Cauchy dataset. The deterministic and randomized leverage score sampling performed similarly, with near zero optimality gap for all values of $s$. Uniform sampling performed poorly, its optimality gap decreasing with increasing $s$. In Figure 1b we summarize the total computation time for calculating the MVCEs. For the sampled datasets, this also includes the computation time for calculating leverage scores (if applicable), and sampling from the dataset. The total computation time for the full dataset was $2\,450$ seconds, that is, just over $40$ minutes. Uniform sampling was faster than the leverage score sampling methods (due to the calculation of the leverage scores), but had very large optimality gaps (see Figure 1a).

In Figure 2a, we summarize the calculated optimality gaps $\tilde{g}^* - \tilde{g}^*_s$ for the Lognormal dataset. The deterministic and randomized leverage score sampling performed similarly, with zero optimality gap for all values of $s$. Uniform sampling performed poorly, its optimality gap decreasing with increasing $s$. In Figure 2b we summarize the total computation time for calculating the MVCEs. The total computation time for the full dataset was $2564$ seconds, that is, just over $42$ minutes. Uniform sampling was generally faster than the leverage score sampling methods (due to the calculation of the leverage scores), but had very large optimality gaps (see Figure 2a).

In Figure 3a, we summarize the calculated optimality gaps $\tilde{g}^* - \tilde{g}^*_s$ for the Gaussian dataset. Only the deterministic leverage score sampling performed well, with near zero optimality gap for all $s$. The uniform and randomized leverage score sampling performed similarly, with optimality gaps decreasing with increasing $s$. This is unsurprising, since the leverage scores for a Gaussian dataset are close to uniform. In Figure 3b we summarize the total computation time for calculating the MVCEs. The total computation time for the full dataset was very large, at $141\,840$ seconds, that is, over $39$ hours. As with the other dataset, uniform sampling was the fastest, but at the cost of a larger optimality gap (see Figure 3a). Additionally, both leverage score sampling methods had similar runtimes, but only the deterministic sampling had near zero optimality gap (see Figure 3a).

Overall, the deterministic leverage score sampling performs the best, achieving both a small optimality gap and greatly decreasing computation time on all three datasets.

# 8 CONCLUSION

In this paper, we have provided the first theoretical guarantees on the quality of initial and final solutions for the MVCE problem, when sampling points deterministically according to their statistical leverage scores. We proved this approach is efficient, assuming the leverage scores exhibit a power law decay. Numerical results show that our data-driven leverage score sampling algorithm performs even better than the established theoretical error bounds, even in cases where the leverage scores are close to uniform distribution, which could be a by-product of our analysis. Future work could include extending these results to other data-driven sampling methods, including randomized leverage score sampling, and also implementing this algorithm extensively on real-world large-scale datasets.

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

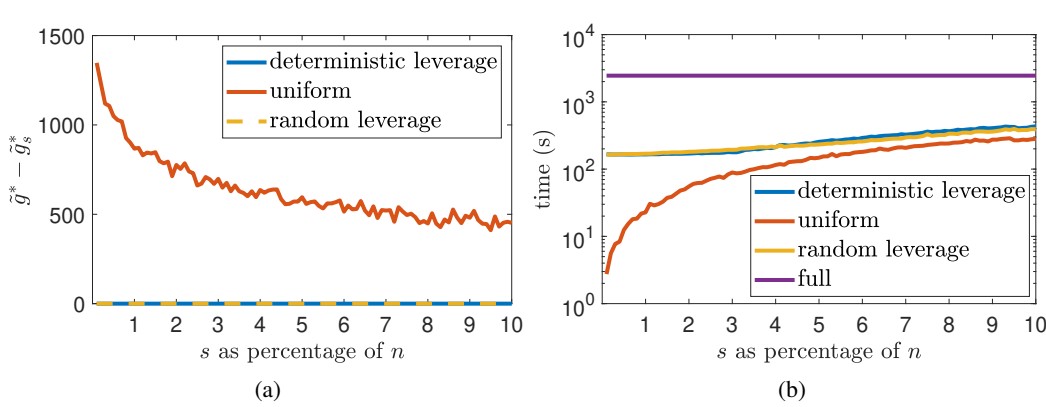

Figure 1: Rotated Cauchy: Calculated optimality gap summary (a) and time summary (b).

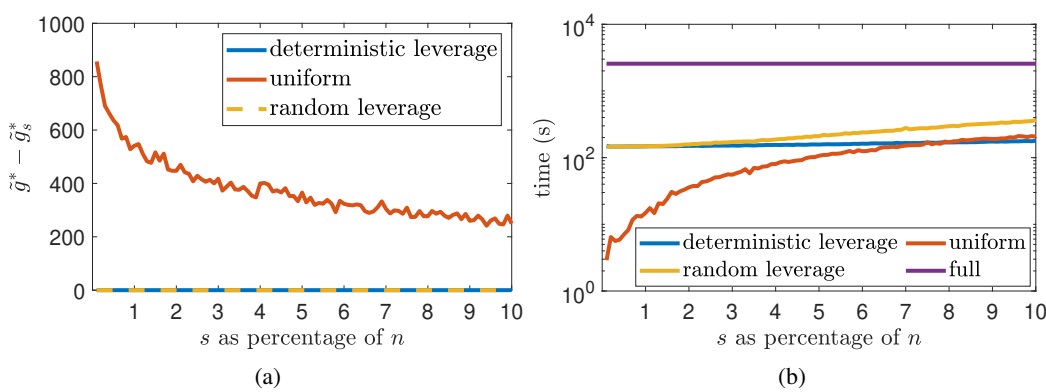

Figure 2: Lognormal: Calculated optimality gap summary (a) and time summary (b).

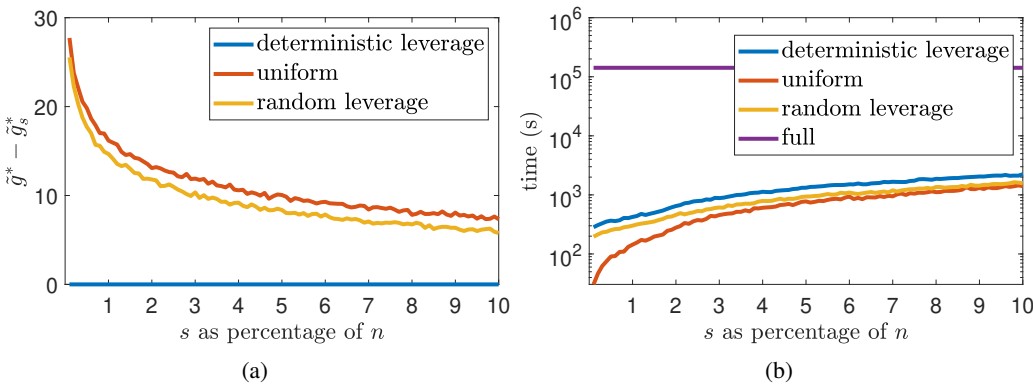

Figure 3: Gaussian: Calculated optimality gap summary (a) and time summary (b).

Yu-Jen Chen, Ming-Yi Ju, and Kao-Shing Hwang. A virtual torque-based approach to kinematic control of redundant manipulators. *IEEE Transactions on Industrial Electronics*, 64(2):1728–1736, 2016. doi: 10.1109/TIE.2016.2548439.

F. L. Chernousko. Ellipsoidal state estimation for dynamical systems. *Nonlinear Analysis: Theory, Methods & Applications*, 63(5-7):872–879, 2005. doi: 10.1016/j.na.2005.01.009.

Michael B. Cohen, Yin Tat Lee, Cameron Musco, Christopher Musco, Richard Peng, and Aaron Sidford. Uniform sampling for matrix approximation. In *Proceedings of the 2015 Conference on Innovations in Theoretical Computer Science*, pp. 181–190, 2015.

Michael B Cohen, Ben Cousins, Yin Tat Lee, and Xin Yang. A near-optimal algorithm for approximating the John ellipsoid. In *Conference on Learning Theory*, pp. 849–873. PMLR, 2019.

S Damla Ahipasaoglu, Peng Sun, and Michael J Todd. Linear convergence of a modified Frank–Wolfe algorithm for computing minimum-volume enclosing ellipsoids. *Optimisation Methods and Software*, 23(1):5–19, 2008.

P. Drineas, M. Magdon-Ismail, M.W. Mahoney, and D.P. Woodruff. Fast approximation of matrix coherence and statistical leverage. *Journal of Machine Learning Research*, 13(Dec):3475–3506, 2012.

Petros Drineas, Michael W. Mahoney, and Shan Muthukrishnan. Relative-error CUR matrix decompositions. *SIAM Journal on Matrix Analysis and Applications*, 30(2):844–881, 2008.

David Eberly. *3D game engine design: a practical approach to real-time computer graphics*. CRC Press, 2006. doi: 10.1201/b18212.

Ali Eshragh, Luke Yerbury, Asef Nazari, Fred Roosta, and Michael W. Mahoney. SALSA: Sequential approximate leverage-score algorithm with application in analyzing big time series data. *arXiv preprint arXiv:2401.00122*, 2023.

Jordi Fonollosa. Gas sensor array under dynamic gas mixtures. UCI Machine Learning Repository, 2015. DOI: https://doi.org/10.24432/C5WP4C. License: CC BY 4.0.

Marguerite Frank, Philip Wolfe, et al. An algorithm for quadratic programming. *Naval research logistics quarterly*, 3(1-2):95–110, 1956. doi: 10.1002/nav.3800030109.

Radoslav Harman, Lenka Filová, and Peter Richtárik. A randomized exchange algorithm for computing optimal approximate designs of experiments. *Journal of the American Statistical Association*, 115(529):348–361, 2020. doi: 10.1080/01621459.2018.1546588.

Fritz John. Extremum problems with inequalities as subsidiary conditions. In *Traces and Emergence of Nonlinear Programming*, pp. 197–215. Springer Basel, 2014. doi: 10.1007/978-3-0348-0439-4_9.

Linus Källberg and Daniel Andrén. Active set strategies for the computation of minimum-volume enclosing ellipsoids. Technical report, Mälardalen Real-Time Research Centre, Mälardalen University, November 2019. URL http://www.es.mdu.se/publications/5680-.

Leonid G. Khachiyan. Rounding of polytopes in the real number model of computation. *Mathematics of Operations Research*, 21(2):307–320, 1996. doi: 10.1287/moor.21.2.307.

Leonid G. Khachiyan and Michael J. Todd. On the complexity of approximating the maximal inscribed ellipsoid for a polytope. *Mathematical Programming*, 61(1-3):137–159, 1993.

Jakub Kudela. Minimum-volume covering ellipsoids: Improving the efficiency of the Wolfe-Atwood algorithm for large-scale instances by pooling and batching. In *MENDEL*, volume 25, pp. 19–26, 2019.

Piyush Kumar and E. Alper Yildirim. Minimum-volume enclosing ellipsoids and core sets. *Journal of Optimization Theory and applications*, 126(1):1–21, 2005. doi: 10.1007/s10957-005-2653-6.

Shannon R McCurdy, Vasilis Ntranos, and Lior Pachter. Deterministic column subset selection for single-cell rna-seq. *Plos one*, 14(1):e0210571, 2019.

Yurii Nesterov and Arkadii Nemirovskii. *Interior-point polynomial algorithms in convex programming*. SIAM, 1994.

Bruno Ordozgoiti, Antonis Matakos, and Aristides Gionis. Generalized leverage scores: Geometric interpretation and applications. In *International Conference on Machine Learning*, pp. 17056–17070. PMLR, 2022.

Dimitris Papailiopoulos, Anastasios Kyrillidis, and Christos Boutsidis. Provable deterministic leverage score sampling. In *Proceedings of the 20th ACM SIGKDD international conference on Knowledge discovery and data mining*, pp. 997–1006, 2014.

Samuel Rosa and Radoslav Harman. Computing minimum-volume enclosing ellipsoids for large datasets. *Computational Statistics & Data Analysis*, 171:107452, 2022. doi: 10.1016/j.csda.2022.107452.

Judah Ben Rosen. Pattern separation by convex programming. *Journal of Mathematical Analysis and Applications*, 10(1):123–134, 1965. doi: 10.1016/0022-247X(65)90150-2.

Peter J. Rousseeuw and Mia Hubert. Robust statistics for outlier detection. *Wiley interdisciplinary reviews: Data mining and knowledge discovery*, 1(1):73–79, 2011. doi: 10.1002/widm.2.

Fred Schweppe. Recursive state estimation: Unknown but bounded errors and system inputs. *IEEE Transactions on Automatic Control*, 13(1):22–28, 1968. doi: 10.1109/TAC.1968.1098790.

R. Sibson. Discussion of Dr Wynn's and of Dr Laycock's papers. *Journal of the Royal Statistical Society: Series B (Methodological)*, 34(2):181–183, 1972. doi: 10.1111/j.2517-6161.1972.tb00898.x.

S. D. Silvey. *Optimal design: and introduction to the theory for parameter estimation*. London, Chapman and Hall/CRC, 1980. doi: 10.1007/978-94-009-5912-5.

Zhao Song, Xin Yang, Yuanyuan Yang, and Tianyi Zhou. Faster algorithm for structured John ellipsoid computation. *arXiv preprint arXiv:2211.14407*, 2022.

Peng Sun and Robert M. Freund. Computation of minimum-volume covering ellipsoids. *Operations Research*, 52(5):690–706, 2004. doi: 10.1287/opre.1040.0115.

D. M. Titterington. Optimal design: some geometrical aspects of D-optimality. *Biometrika*, 62(2):313–320, 1975. doi: 10.2307/2335366.

D. M. Titterington. Estimation of correlation coefficients by ellipsoidal trimming. *Journal of the Royal Statistical Society: Series C (Applied Statistics)*, 27(3):227–234, 1978. doi: 10.2307/2347157.

Michael J. Todd. *Minimum-volume ellipsoids: Theory and algorithms*. SIAM, 2016a. doi: 10.1137/1.9781611974386.

Michael J. Todd. Minvol. SIAM, 2016b. URL: http://archive.siam.org/books/mo23/.

Michael J. Todd and E. Alper Yıldırım. On Khachiyan's algorithm for the computation of minimum-volume enclosing ellipsoids. *Discrete Applied Mathematics*, 155(13):1731–1744, 2007. doi: 10.1016/j.dam.2007.02.013.

Philip Wolfe. Convergence theory in nonlinear programming. *Integer and nonlinear programming*, pp. 1–36, 1970. doi: 10.1007/BF00932858.

David P Woodruff and Taisuke Yasuda. New subset selection algorithms for low rank approximation: Offline and online. In *Proceedings of the 55th Annual ACM Symposium on Theory of Computing*, pp. 1802–1813, 2023.

David P. Woodruff and Taisuke Yasuda. John ellipsoids via lazy updates. *arXiv preprint arXiv:2501.01801*, 2025.

Yaming Yu. D-optimal designs via a cocktail algorithm. *Statistics and Computing*, 21(4):475–481, 2011. doi: 10.1007/s11222-010-9183-2.

## A  DETERMINISTIC APPROXIMATE LEVERAGE SCORE SAMPLING

The computation time in Algorithm 1 is dominated by the cost of calculating the leverage scores exactly, which is $\mathcal{O}\left(nd^2\right)$. We can instead use approximate leverage scores, which are more computationally efficient to calculate. We demonstrate the approach using the algorithm of Drineas et al. (2012).

**Theorem 6** (Drineas et al. (2012), Theorem 2). *Fix a constant $\beta \in (0, 1/2]$. Let $X \in \mathbb{R}^{n \times d}$ be full rank, with $n \gg d$. Let the leverage scores of the rows of $X$ be given by $\ell_1(X), \ldots, \ell_n(X)$. Then there exists a randomized algorithm that calculates approximate leverage scores $\hat{\ell}_1(X), \ldots, \hat{\ell}_n(X)$ such that with probability at least 0.8, simultaneously for all $i = 1, \ldots, n$, $\hat{\ell}_i(X) = (1 \pm \beta) \ell_i(X)$. This algorithm has time complexity $\mathcal{O}(nd \log n)$.*

We then sample deterministically according to these approximate leverage scores. We summarize the modified sampling procedure in Algorithm 2.

---

**Algorithm 2** Deterministic Approximate Leverage Score Sampling with Threshold

---

**Require:** $X = [x_1, \ldots, x_n]^\mathsf{T} \in \mathbb{R}^{n \times d}, \beta \in (0, 1/2], \varepsilon \in (0, 1)$
1: Calculate approximate leverage scores for each row in $X$, using Theorem 6. Assume $\hat{\ell}_1(X) \geq \cdots \geq \hat{\ell}_n(X)$.
2: Let $t = \sum_{i=1}^{n} \hat{\ell}_i(X)$.
3: Let $s = \arg\min_j \left( \sum_{i=1}^{j} \hat{\ell}_i(X) \geq t - (1-\beta)\varepsilon \right)$.
4: Let $R = 0 \in \mathbb{R}^{s \times n}$.
5: **for** $i = 1 : s$ **do**
6:     Set row $i$ of $R$ equal to $e_i$.
7: **end for**
**Ensure:** $R, s$

---

We have chosen $s$ carefully, to ensure the following subspace embedding result.

**Corollary 7.** *Let $\beta \in (0, 1/2]$, and $\varepsilon \in (0, 1)$. Use Algorithm 2 to construct $X_s = RX$. Then, with probability at least 0.8, we have*

$$(1 - \varepsilon)X^\mathsf{T}X \prec X_s^\mathsf{T}X_s \preceq X^\mathsf{T}X.$$

*Proof.* For this proof, we only need to show that $\sum_{i=s+1}^{n} \ell_i(X) < \varepsilon$ holds with probability at least 0.8. We can then apply Theorem 2 to obtain our bound.

Assume that $\hat{\ell}_i(X) \geq (1-\beta)\ell_i(X)$ holds simultaneously for all $i = 1, \ldots, n$. By construction of Algorithm 2, we have

$$\sum_{i=s+1}^{n} \hat{\ell}_i(X) < (1-\beta)\varepsilon.$$

Then since $\hat{\ell}_i(X) \geq (1-\beta)\ell_i(X)$, for all $i = 1, \ldots, n$, we have

$$(1-\beta)\varepsilon > \sum_{i=s+1}^{n} \hat{\ell}_i(X) \geq \sum_{i=s+1}^{n} (1-\beta)\ell_i(X) = (1-\beta)\sum_{i=s+1}^{n} \ell_i(X),$$

that is,

$$\sum_{i=s+1}^{n} \ell_i(X) < \varepsilon.$$

$\square$

**Corollary 8.** *If the leverage scores exhibit a power law decay, then $s = poly\left(d(1+\beta), \frac{1}{(1-\beta)\varepsilon}\right)$.*

*Proof.* Papailiopoulos et al. (2014) show that if the leverage scores exhibit a power law decay, then Algorithm 1 outputs

$$s = \arg\min_j \left( \sum_{i=1}^j \ell_i \left( \boldsymbol{X} \right) > d - \varepsilon \right) = \max \left\{ \left( \frac{2d}{\varepsilon} \right)^{\frac{1}{1+\eta}} - 1, \left( \frac{2d}{\eta\varepsilon} \right)^{\frac{1}{\eta}} - 1, d \right\} = \text{poly} \left( d, \frac{1}{\varepsilon} \right).$$

Then, comparing Algorithm 1 with Algorithm 2, we need only replace $d$ with $t$, and $\varepsilon$ with $(1 - \beta)\varepsilon$. That is, Algorithm 2 outputs

$$s = \arg\min_j \left( \sum_{i=1}^j \hat{\ell}_i \left( \boldsymbol{X} \right) > t - (1 - \beta)\varepsilon \right) = \text{poly} \left( t, \frac{1}{(1 - \beta)\varepsilon} \right).$$

But

$$t = \sum_{i=1}^n \hat{\ell}_i \left( \boldsymbol{X} \right) \le \sum_{i=1}^n (1 + \beta) \ell_i \left( \boldsymbol{X} \right) = (1 + \beta) \sum_{i=1}^n \ell_i \left( \boldsymbol{X} \right) = (1 + \beta) d.$$

Hence $s = \text{poly} \left( d(1 + \beta), \frac{1}{(1-\beta)\varepsilon} \right)$. □

## B   REPEATED APPLICATIONS OF LEVERAGE SCORE SAMPLING

For many datasets, a single application of Algorithm 1 does not sufficiently reduce the number of points considered by the MVCE algorithm. We show how multiple applications of Algorithm 1 can still guarantee a subspace embedding result. We summarize the modified sampling procedure in Algorithm 3.

---

**Algorithm 3** Repeated Deterministic Leverage Score Sampling with Threshold

---

**Require:** $\boldsymbol{X} = [\boldsymbol{x}_1, \ldots, \boldsymbol{x}_n]^\mathsf{T} \in \mathbb{R}^{n \times d}, \varepsilon \in (0, 1), m \in \mathbb{N}$
1: Let $\boldsymbol{X}_{(0)} = \boldsymbol{X}$.
2: **for** $k = 1 : m$ **do**
3:    Calculate leverage scores for each row in $\boldsymbol{X}_{(k)}$. Assume $\ell_1 \left( \boldsymbol{X}_{(k)} \right) \ge \cdots \ge \ell_n \left( \boldsymbol{X}_{(k)} \right)$.
4:    Let $s = \arg\min_j \left( \sum_{i=1}^j \ell_i \left( \boldsymbol{X}_{(k)} \right) > d - \varepsilon \right)$.
5:    Let $\boldsymbol{R}_{(k)} = \boldsymbol{0} \in \mathbb{R}^{s \times n}$.
6:    **for** $i = 1 : s$ **do**
7:       Set row $i$ of $\boldsymbol{R}_{(k)}$ equal to $\boldsymbol{e}_i$.
8:    **end for**
9:    Let $\boldsymbol{X}_{(k+1)} = \boldsymbol{R}_{(k)} \boldsymbol{X}_{(k)}$.
10:   **if** $\text{rank} \left( \boldsymbol{X}_{(k+1)} \right) < d$ **then**
11:      Redefine $m := k$.
12:      **break**
13:   **end if**
14: **end for**
**Ensure:** $\boldsymbol{X}_s = \boldsymbol{X}_{(m)}$

---

We now present the subspace embedding result.

**Corollary 9.** *Let $\varepsilon \in (0, 1)$, and $m \in \mathbb{N}$. Use Algorithm 3 to construct $\boldsymbol{X}_s$. Then*

$$(1 - \varepsilon)^m \boldsymbol{X}^\mathsf{T} \boldsymbol{X} \prec \boldsymbol{X}_s^\mathsf{T} \boldsymbol{X}_s.$$

*Proof.* The proof relies on the repeated application of Theorem 2. Essentially, we prove

$$(1 - \varepsilon)^m \boldsymbol{X}_{(0)}^\mathsf{T} \boldsymbol{X}_{(0)} \prec (1 - \varepsilon)^{m-1} \boldsymbol{X}_{(1)}^\mathsf{T} \boldsymbol{X}_{(1)} \prec \cdots \prec (1 - \varepsilon) \boldsymbol{X}_{(m-1)}^\mathsf{T} \boldsymbol{X}_{(m-1)} \prec \boldsymbol{X}_{(m)}^\mathsf{T} \boldsymbol{X}_{(m)}, \tag{7}$$

which we can show by induction. When $m = 1$, Theorem 2 implies

$$(1 - \varepsilon)^1 \boldsymbol{X}_{(0)}^\mathsf{T} \boldsymbol{X}_{(0)} = (1 - \varepsilon) \boldsymbol{X}^\mathsf{T} \boldsymbol{X} \prec \boldsymbol{X}_s^\mathsf{T} \boldsymbol{X}_s = \boldsymbol{X}_{(1)}^\mathsf{T} \boldsymbol{X}_{(1)},$$

proving the base case. Now suppose (7) holds for some $m = k > 1$. That is, assume

$$(1 - \varepsilon)^k \boldsymbol{X}_{(0)}^\mathsf{T} \boldsymbol{X}_{(0)} \prec \cdots \prec \boldsymbol{X}_{(k)}^\mathsf{T} \boldsymbol{X}_{(k)}. \tag{8}$$

That is, we have

$$(1 - \varepsilon)^{k+1} \boldsymbol{X}_{(0)}^\mathsf{T} \boldsymbol{X}_{(0)} \prec \cdots \prec (1 - \varepsilon) \boldsymbol{X}_{(k)}^\mathsf{T} \boldsymbol{X}_{(k)}. \tag{9}$$

Additionally, if we sample from $\boldsymbol{X}_{(k)}$ using Algorithm 1, Theorem 2 guarantees that

$$(1 - \varepsilon) \boldsymbol{X}_{(k)}^\mathsf{T} \boldsymbol{X}_{(k)} \prec \boldsymbol{X}_{(k+1)}^\mathsf{T} \boldsymbol{X}_{(k+1)}. \tag{10}$$

Combining (9) and (10), we obtain

$$(1 - \varepsilon)^{k+1} \boldsymbol{X}_{(0)}^\mathsf{T} \boldsymbol{X}_{(0)} \prec \cdots \prec (1 - \varepsilon) \boldsymbol{X}_{(k)}^\mathsf{T} \boldsymbol{X}_{(k)} \prec \boldsymbol{X}_{(k+1)}^\mathsf{T} \boldsymbol{X}_{(k+1)},$$

finalizing the proof. $\qquad\square$

## C   MISSING DETAILS IN PROOF OF THEOREM 3

PROOF OF EQUATIONS (1) AND (2)

We have that

$$
\begin{aligned}
g(\boldsymbol{u}_K) &= \log \det \left( \boldsymbol{X}^\mathsf{T} \boldsymbol{U}_K \boldsymbol{X} \right) \\
&= \log \det \left( \frac{1}{n} \boldsymbol{X}^\mathsf{T} \boldsymbol{X} \right) \\
&= \log \left( \frac{1}{n^d} \det \left( \boldsymbol{X}^\mathsf{T} \boldsymbol{X} \right) \right) \\
&= \log \det \left( \boldsymbol{X}^\mathsf{T} \boldsymbol{X} \right) - d \log n,
\end{aligned}
$$

and, similarly,

$$
\begin{aligned}
g_s(\boldsymbol{u}_0) &= \log \det \left( \boldsymbol{X}_s^\mathsf{T} \boldsymbol{U}_0 \boldsymbol{X}_s \right) \\
&= \log \det \left( \frac{1}{s} \boldsymbol{X}_s^\mathsf{T} \boldsymbol{X}_s \right) \\
&= \log \det \left( \boldsymbol{X}_s^\mathsf{T} \boldsymbol{X}_s \right) - d \log s.
\end{aligned}
$$

PROOF OF INEQUALITY (4)

We first need to prove the following lemma.

**Lemma 10.** *Suppose we have two $d \times d$ symmetric positive semidefinite matrices $\boldsymbol{A}$ and $\boldsymbol{B}$, with $\boldsymbol{A} \preceq \boldsymbol{B}$. Let the eigenvalues of $\boldsymbol{A}$ be given by $\lambda_1 (\boldsymbol{A}) \leq \cdots \leq \lambda_d (\boldsymbol{A})$, and the eigenvalues of $\boldsymbol{B}$ be given by $\lambda_1 (\boldsymbol{B}) \leq \cdots \leq \lambda_d (\boldsymbol{B})$. Then for all $i = 1, \ldots, d$, we have*

$$\lambda_i (\boldsymbol{A}) \leq \lambda_i (\boldsymbol{B}).$$

*Proof.* Use Min-max Theorem, along with the fact that $\boldsymbol{x}^\mathsf{T} \boldsymbol{A} \boldsymbol{x} \leq \boldsymbol{x}^\mathsf{T} \boldsymbol{B} \boldsymbol{x}$ for all $\boldsymbol{x}$. $\qquad\square$

We can now prove Inequality (4).

**Lemma 11.** *Suppose we have two $d \times d$ symmetric positive semidefinite matrices $\boldsymbol{A}$ and $\boldsymbol{B}$, with $\boldsymbol{A} \preceq \boldsymbol{B}$. Then*

$$\det (\boldsymbol{A}) \leq \det (\boldsymbol{B}).$$

*Proof.* We have that

$$\log \det (\boldsymbol{A}) = \log \left( \prod_{i=1}^{d} \lambda_i (\boldsymbol{A}) \right)$$

$$= \sum_{i=1}^{d} \log \lambda_i (\boldsymbol{A})$$

$$\leq \sum_{i=1}^{d} \log \lambda_i (\boldsymbol{B})$$

$$= \log \left( \prod_{i=1}^{d} \lambda_i (\boldsymbol{B}) \right)$$

$$= \log \det (\boldsymbol{B}),$$

where the inequality uses Lemma 10 and the fact that $\log$ is a monotonically increasing function. Then

$$\det(\boldsymbol{A}) = e^{\log \det(\boldsymbol{A})} \leq e^{\log \det(\boldsymbol{B})} = \det(\boldsymbol{B}),$$

where the inequality is because the exponential function is monotonically increasing. $\square$

## D  FINAL OPTIMALITY GAP

We break this proof into two cases:

1. $u_i^* > 0$, for all $i = 1, \ldots, s$, and
2. $u_i^* \geq 0$, for all $i = 1, \ldots, s$.

We do this because the proof of Inequality (6) relies on scaling $\sqrt{U^*}$ so that the first $s$ diagonal terms are $\geq 1$, and the last $n - s$ diagonal terms are $\leq 1$. Having any zeros in the first $s$ terms prevents such a scaling.

We consider Case 1 first. First, follow the proof outline from Section 6. Then, what remains to be shown is that Inequality (6) holds.

CASE 1: $u_i^* > 0$, FOR ALL $i = 1, \ldots, s$

The following lemma will be useful for proving Inequality (6). It shows the change in leverage scores when one row of $\boldsymbol{X}$ is scaled.

**Lemma 12** (Ordozgoiti et al. (2022), Lemma 3.2). *Let $\boldsymbol{X} \in \mathbb{R}^{n \times d}$ have rank $d$. Let $\alpha \geq 0$. Define $\boldsymbol{\Sigma}^{(i)}$ as the diagonal matrix satisfying $\boldsymbol{\Sigma}_{ii}^{(i)} = \alpha$, $\boldsymbol{\Sigma}_{jj}^{(i)} = 1$, if $j \neq i$. Then*

$$\ell_j \left( \boldsymbol{\Sigma}^{(i)} \boldsymbol{X} \right) = \ell_j (\boldsymbol{X}) - \frac{(\alpha^2 - 1) \left( \boldsymbol{x}_j^\mathsf{T} \left( \boldsymbol{X}^\mathsf{T} \boldsymbol{X} \right)^\mathsf{T} \boldsymbol{x}_i \right)^2}{1 + (\alpha^2 - 1) \ell_i (\boldsymbol{X})},$$

*and, in particular,*

$$\ell_i \left( \boldsymbol{\Sigma}^{(i)} \boldsymbol{X} \right) = \frac{\alpha^2 \ell_i (\boldsymbol{X})}{1 + (\alpha^2 - 1) \ell_i (\boldsymbol{X})}.$$

We now ready to prove the inequality.

**Proposition 13.** *Use Algorithm 1 to construct $\boldsymbol{X}_s = \boldsymbol{R}\boldsymbol{X}$, with $\varepsilon \in (0, 1)$. Suppose that an optimal solution $\boldsymbol{u}^*$ of (D) satisfies $u_i^* > 0$ for all $i = 1, \ldots, s$. Define $\boldsymbol{Y} := \sqrt{\boldsymbol{U}^*} \boldsymbol{X}$. Then*

$$\sum_{i=s+1}^{n} \ell_i (\boldsymbol{Y}) \leq \sum_{i=s+1}^{n} \ell_i (\boldsymbol{X}).$$

*Proof.* The structure of this proof is based on the proof of Theorem 3.2 in Ordozgoiti et al. (2022). We will show that the sum of the last $n-s$ leverage scores of $\boldsymbol{X}$ do not increase when we premultiply it by $\sqrt{\boldsymbol{U}^*}$. Without loss of generality, we assume that $\min_{i=1,\dots,s} \sqrt{u_i^*} \geq \max_{i=s+1,\dots,n} \sqrt{u_i^*}$. This is because, for any $k$, we can write

$$u_k^* \boldsymbol{x}_k \boldsymbol{x}_k^\mathsf{T} = \sum_{i=1}^\beta \frac{u_k^*}{\beta} \boldsymbol{x}_k \boldsymbol{x}_k^\mathsf{T},$$

where $\beta \in \mathbb{N}$, such that $\frac{u_k^*}{\beta}$ is sufficiently small.

First, we scale $\sqrt{\boldsymbol{U}^*}$ so that the first $s$ diagonal terms are $\geq 1$, and the last $n-s$ diagonal terms are $\leq 1$. Let $\alpha = 1/\min_{i=1,\dots,s} \sqrt{u_i^*}$. We define

$$\boldsymbol{V} := \alpha \sqrt{\boldsymbol{U}^*}.$$

We exploit the fact that

$$\begin{aligned}
\ell_i(\boldsymbol{VX}) &= \ell_i\left(\alpha\sqrt{\boldsymbol{U}^*}\boldsymbol{X}\right) \\
&= \ell_i(\alpha\boldsymbol{Y}) \\
&= \left((\alpha\boldsymbol{Y})\left((\alpha\boldsymbol{Y})^\mathsf{T}(\alpha\boldsymbol{Y})\right)^{-1}(\alpha\boldsymbol{Y})^\mathsf{T}\right)_{ii} \\
&= \left(\alpha^2 \frac{1}{\alpha^2}\boldsymbol{Y}\left(\boldsymbol{Y}^\mathsf{T}\boldsymbol{Y}\right)^{-1}\boldsymbol{Y}^\mathsf{T}\right)_{ii} \\
&= \ell_i(\boldsymbol{Y}).
\end{aligned}$$

To see how $\boldsymbol{V}$ affects the leverage scores of $\boldsymbol{X}$, we will consider scaling each row separately. We define $\boldsymbol{V}^{(i)}$ as the diagonal matrix satisfying $\boldsymbol{V}_{ii}^{(i)} = \boldsymbol{V}_{ii} = \alpha\sqrt{u_i^*}$ and $\boldsymbol{V}_{jj}^{(i)} = 1$, if $j \neq i$. Now, we consider the leverage scores of $\boldsymbol{V}^{(i)}\boldsymbol{X}$. From Lemma 12, we have

1. If $i \leq s$, then $\alpha\sqrt{u_i^*} \geq 1$. Hence $\ell_i\left(\boldsymbol{V}^{(i)}\boldsymbol{X}\right) \geq \ell_i(\boldsymbol{X})$, and $\ell_j\left(\boldsymbol{V}^{(i)}\boldsymbol{X}\right) \leq \ell_j(\boldsymbol{X})$, if $j \neq i$.

2. If $i \geq s$, then $\alpha\sqrt{u_i^*} \leq 1$. Hence $\ell_i\left(\boldsymbol{V}^{(i)}\boldsymbol{X}\right) \leq \ell_i(\boldsymbol{X})$, and $\ell_j\left(\boldsymbol{V}^{(i)}\boldsymbol{X}\right) \geq \ell_j(\boldsymbol{X})$, if $j \neq i$.

Now, we consider scaling the first $s$ rows. From the previous discussion, we can conclude that the leverage scores of the last $n-s$ rows do not increase. Therefore, the sum of the first $s$ leverage scores does not decrease. (This is because the sum of the leverage scores remains constant, since scaling rows by non-zero constants does not affect the rank.) Next, consider scaling the last $n-s$ rows. From the discussion above, all the leverage scores of the first $s$ rows cannot decrease. That is, the sum of the first $s$ leverage scores again does not decrease. We conclude that the sum of the last $n-s$ leverage scores cannot increase. (This is because the sum of the leverage scores remains constant. Suppose rank$(\boldsymbol{VX}) < d$. Then rank$\left(\boldsymbol{X}^\mathsf{T}\boldsymbol{U}^*\boldsymbol{X}\right) = $ rank$\left((\boldsymbol{VX})^\mathsf{T}(\boldsymbol{VX})\right) = $ rank$(\boldsymbol{VX}) < d$, which is a contradiction, since $\boldsymbol{X}^\mathsf{T}\boldsymbol{U}^*\boldsymbol{X}$ is invertible.) Hence

$$\sum_{i=1}^n \ell_i(\boldsymbol{Y}) = \sum_{i=1}^n \ell_i(\boldsymbol{VX}) \leq \sum_{i=1}^n \ell_i(\boldsymbol{X}).$$

$\square$

CASE 2: $u_i^* \geq 0$, FOR ALL $i = 1, \dots, s$

This time, we will examine a $\delta$-feasible solution $\boldsymbol{u}$ for (D), which has $u_i > 0$ for all $i = 1, \dots, n$. Such a solution can be achieved by using the Frank-Wolfe algorithm (Frank et al., 1956; Wolfe, 1970) with Khachiyan's initialisation (Khachiyan, 1996).

Consider the feasible solution $\tilde{\boldsymbol{u}}_s$ for (D$_s$), given by

$$\tilde{\boldsymbol{u}}_s = \frac{1}{\boldsymbol{e}^\top \boldsymbol{u}_s} \boldsymbol{u}_s,$$

where $\boldsymbol{u}_s$ contains the first $s$ entries of $\boldsymbol{u}$. We would like a bound similar to the one in Theorem 2, with $\boldsymbol{X}$ replaced with

$$\boldsymbol{Y} := \sqrt{\boldsymbol{U}} \boldsymbol{X}.$$

We require the following result.

**Corollary 14.** *Use Algorithm 1 to construct $\boldsymbol{X}_s = \boldsymbol{R}\boldsymbol{X}$, with $\varepsilon \in (0,1)$. Let $\boldsymbol{u}$ be a $\delta$-feasible solution to (D). Then*

$$\sum_{i=s+1}^{n} \ell_i (\boldsymbol{Y}) \leq \sum_{i=s+1}^{n} \ell_i (\boldsymbol{X}).$$

*Proof.* Follow the proof of Proposition 13, but replace $\boldsymbol{u}^*$ with $\boldsymbol{u}$, and redefine $\boldsymbol{Y} := \sqrt{\boldsymbol{U}} \boldsymbol{X}$. $\qquad\square$

Then, by the construction of Algorithm 1, we have

$$\sum_{i=s+1}^{n} \ell_i (\boldsymbol{Y}) \leq \sum_{i=s+1}^{n} \ell_i (\boldsymbol{X}) < \varepsilon,$$

for some $\varepsilon \in (0,1)$. Therefore, we may apply Theorem 2 with $\boldsymbol{Y}$ instead of $\boldsymbol{X}$, to obtain the bound

$$(1-\varepsilon)\boldsymbol{X}^\top \boldsymbol{U}\boldsymbol{X} \prec \boldsymbol{X}_s^\top \boldsymbol{U}_s \boldsymbol{X}_s.$$

That is, the feasible solution $\tilde{\boldsymbol{u}}_s$ satisfies

$$(1-\varepsilon)\boldsymbol{X}^\top \boldsymbol{U}\boldsymbol{X} \preceq \frac{1-\varepsilon}{\boldsymbol{e}^\top \boldsymbol{u}_s} \boldsymbol{X}^\top \boldsymbol{U}_s \boldsymbol{X} \prec \frac{1}{\boldsymbol{e}^\top \boldsymbol{u}_s} \boldsymbol{X}_s^\top \boldsymbol{U}_s \boldsymbol{X}_s = \boldsymbol{X}_s^\top \tilde{\boldsymbol{U}}_s \boldsymbol{X}_s.$$

We are now ready to prove the final optimality gap.

**Theorem 15.** *Use Algorithm 1 to construct $\boldsymbol{X}_s = \boldsymbol{R}\boldsymbol{X}$, with $\varepsilon \in (0,1)$. Let $\boldsymbol{u}$ be a $\delta$-feasible solution to (D). Then*

$$g^* - g_s^* < d \log \left( \frac{1+\delta}{1-\varepsilon} \right).$$

*Proof.* For this proof, we will exploit the fact that

$$g^* - g_s^* = (g^* - g(\boldsymbol{u})) + (g(\boldsymbol{u}) - g_s^*).$$

We will bound above $g(\boldsymbol{u}) - g_s^*$ first. Let $\tilde{\boldsymbol{u}}_s = \frac{1}{\boldsymbol{e}^\top \boldsymbol{u}_s} \boldsymbol{u}_s$, where $\boldsymbol{u}_s$ contains the first $s$ entries of $\boldsymbol{u}$. Since $g_s$ is concave, the optimal value $g_s^*$ must be greater or equal to $g_s$ at any feasible point for (D$_s$). Hence

$$
\begin{aligned}
g_s^* &\geq g_s(\tilde{\boldsymbol{u}}_s) \\
&= \log \det \left( \boldsymbol{X}_s^\top \tilde{\boldsymbol{U}}_s \boldsymbol{X}_s \right) \\
&> \log \det \left( (1-\varepsilon)\boldsymbol{X}^\top \boldsymbol{U}\boldsymbol{X} \right) \\
&= d \log (1-\varepsilon) + \log \det \left( \boldsymbol{X}^\top \boldsymbol{U}\boldsymbol{X} \right),
\end{aligned}
$$

where the inequality uses Lemma 11. Hence

$$
\begin{aligned}
g(\boldsymbol{u}) - g_s^* &< \log \det \left( \boldsymbol{X}^\top \boldsymbol{U}\boldsymbol{X} \right) - \left( d \log (1-\varepsilon) + \log \det \left( \boldsymbol{X}^\top \boldsymbol{U}\boldsymbol{X} \right) \right) \\
&= d \log \left( \frac{1}{1-\varepsilon} \right).
\end{aligned}
$$

Now, recall that $\boldsymbol{u}$ is $\delta$-feasible for (D). By Proposition 1, we have

$$g^* - g(\boldsymbol{u}) \leq d \log \left( 1 + \delta \right).$$

Hence

$$\begin{aligned}
g^* - g_s^* &= (g^* - g(\boldsymbol{u})) + (g(\boldsymbol{u}) - g_s^*) \\
&< d \log \left( \frac{1}{1 - \varepsilon} \right) + d \log \left( 1 + \delta \right) \\
&= d \log \left( \frac{1 + \delta}{1 - \varepsilon} \right).
\end{aligned}$$

$\square$

# E    MORE NUMERICAL RESULTS

## COMPARISON OF OUR ALGORITHM WITH THE FIXED POINT ALGORITHM

All computations in this section are performed on a personal laptop with a 64 bit Windows 11 Home operating system (Version 23H2), and a 3.30 GHz 11th Gen Intel Core i7-11370H processor with 40 GB of RAM. The algorithms are run using MATLAB (R2021a).

In this subsection, we compare our algorithm to the Fixed Point algorithm (Cohen et al., 2019). For consistency, we run our algorithm using $s = 10\%$ of $n$. We run both algorithms on Gaussian datasets of size $n = 1\,000\,000$, with dimension $d$ ranging from 2 to 10. Table 1 summarizes the total computation time required for both algorithms, for varying values of accuracy parameter $\delta$.

Table 1: Time summary for the two algorithms. Tests run on Gaussian datasets of size $n = 1\,000\,000$, and $d$ ranging from 2 to 10. Each value represents mean and sample standard deviation of 100 runs.

| | | | Time (s) | |
|---|---|---|---|---|
| $n$ | $d$ | $\delta$ | Our algorithm | Fixed Point algorithm |
| $1\,000\,000$ | 2 | $10^{-2}$ | $0.0914 \pm 0.0220$ | $1.4331 \pm 0.0877$ |
| | | $10^{-3}$ | $0.0922 \pm 0.0210$ | $12.1627 \pm 0.8764$ |
| | | $10^{-4}$ | $0.0873 \pm 0.0215$ | $63.1388 \pm 6.2418$ |
| | 5 | $10^{-2}$ | $0.1330 \pm 0.0219$ | $3.6558 \pm 0.2074$ |
| | | $10^{-3}$ | $0.1956 \pm 0.0442$ | $12.1783 \pm 1.3380$ |
| | | $10^{-4}$ | $0.2125 \pm 0.0390$ | $47.2873 \pm 11.6625$ |
| | 10 | $10^{-2}$ | $0.2483 \pm 0.0618$ | $6.0298 \pm 0.7460$ |
| | | $10^{-3}$ | $0.4177 \pm 0.1025$ | $21.8280 \pm 3.9056$ |
| | | $10^{-4}$ | $0.4313 \pm 0.1077$ | $117.9867 \pm 41.5669$ |

Table 2: Time summary for the two algorithms. Tests run on Gaussian datasets of dimension $d = 100$, with varying $n$. Each value represents mean and sample standard deviation (when multiple runs are computed).

| | | | | Time (s) | |
|---|---|---|---|---|---|
| $d$ | $n$ | $\delta$ | # runs | Our algorithm | Fixed Point algorithm |
| 100 | $10\,000$ | $10^{-2}$ | 100 | $0.0927 \pm 0.0402$ | $0.7270 \pm 0.1886$ |
| | | $10^{-3}$ | 100 | $0.2578 \pm 0.1991$ | $5.6216 \pm 1.5382$ |
| | | $10^{-4}$ | 100 | $0.3013 \pm 0.0833$ | $18.6236 \pm 4.8307$ |
| | | $10^{-5}$ | 50 | $0.5162 \pm 0.1292$ | $82.4097 \pm 26.7802$ |
| | $100\,000$ | $10^{-3}$ | 20 | $1.9359 \pm 0.1371$ | $160.1398 \pm 34.8062$ |
| | $1\,000\,000$ | $10^{-3}$ | 1 | $101.0625$ | $4837.5313$ |

We now apply the same methods to Gaussian matrices of dimension $d = 100$, with varying number of points $n$ and accuracy parameter $\delta$. Table 2 summarizes the total computation time required for both algorithms.

In both Tables 1 and 2, we notice that as $\delta$ increases, the time taken for both methods generally increases. However, the increase in time for our algorithm is minimal, while the increase in time for the Fixed Point algorithm is significant. In the final row of Table 2, our algorithm takes less than 2 minutes, compared with the Fixed Point algorithm, which took about 80 minutes.

We emphasize that the examples considered here are smaller than those we considered in Section 7, and the accuracy parameter $\delta$ is much larger than our desired $10^{-9}$. Regardless, we ran the Fixed Point algorithm on the examples from Section 7. We found that for all three datasets, the algorithm did not converge in two hours of runtime.

REAL WORLD DATA

All computations in this section are performed on a personal laptop with a 64 bit MacOS 13 operating system, and a 2.4 GHz Quad-Core Intel Core i5 processor with 8 GB of RAM. The algorithms are run using MATLAB (R2021a).

We now test our algorithm on three smaller datasets from the UCI Machine Learning Repository. Table 3 summarizes the size of each dataset, and time to compute the MVCE using the WA algorithm (Wolfe, 1970; Atwood, 1973).

We then apply our algorithm on each dataset, varying sample size $s$ from $1\%$ to $10\%$ of $n$. In Table 4, we summarize the optimality gaps. The deterministic sampling performed well on all datasets, achieving very small optimality gaps when $s$ is $10\%$ of $n$. In Table 5, we summarize the computation times. Although the deterministic sampling was the slowest, all sampling methods decreased the total computation time required to calculate the MVCE.

Table 3: Dataset description. This includes dimension and time to compute MVCE using the WA algorithm.

| Dataset | $n$ | $d$ | Time (s) |
|---|---|---|---|
| Ethylene CO (Fonollosa, 2015) | 4 208 261 | 19 | $85.92 \pm 0.17$ |
| Ethylene CH4 (Fonollosa, 2015) | 4 178 504 | 19 | $75.89 \pm 0.38$ |
| Skin (Bhatt & Dhall, 2012) | 245 057 | 4 | $0.54 \pm 0.04$ |

Table 4: Optimality gap summary. Sample size $s$ takes values of 1%, 5%, and 10% of $n$. Each value represents mean and sample standard deviation of 5 runs.

| Dataset | $s$ | det | Optimality Gap prob | unif |
|---|---|---|---|---|
| Ethylene CO (Fonollosa, 2015) | 1% | 4.66 | $1.01 \pm 0.11$ | $5.47 \pm 0.65$ |
| | 5% | 0.02 | $0.43 \pm 0.03$ | $4.40 \pm 1.03$ |
| | 10% | 5.67E-06 | $0.25 \pm 0.04$ | $3.24 \pm 0.48$ |
| Ethylene CH4 (Fonollosa, 2015) | 1% | 1.59 | $1.48 \pm 0.09$ | $3.53 \pm 0.42$ |
| | 5% | 0.05 | $0.66 \pm 0.05$ | $1.69 \pm 0.15$ |
| | 10% | 0.01 | $0.37 \pm 0.08$ | $1.35 \pm 0.11$ |
| Skin (Bhatt & Dhall, 2012) | 1% | 0.75 | $0.07 \pm 0.02$ | $0.25 \pm 0.13$ |
| | 5% | 0.56 | $0.02 \pm 0.01$ | $0.05 \pm 0.03$ |
| | 10% | -3.55E-15 | $0.03 \pm 0.01$ | $0.04 \pm 0.02$ |

Table 5: Time summary. Sample size $s$ takes values of 1%, 5%, and 10% of $n$. Each value represents mean and sample standard deviation of 5 runs.

| Dataset | $s$ | det | Time (s) prob | unif |
|---|---|---|---|---|
| Ethylene CO (Fonollosa, 2015) | 1% | $6.00 \pm 0.12$ | $5.42 \pm 0.74$ | $4.01 \pm 0.39$ |
| | 5% | $15.00 \pm 0.56$ | $9.94 \pm 0.47$ | $7.55 \pm 0.05$ |
| | 10% | $15.71 \pm 0.14$ | $14.32 \pm 1.26$ | $11.66 \pm 0.95$ |
| Ethylene CH4 (Fonollosa, 2015) | 1% | $5.91 \pm 0.30$ | $5.24 \pm 0.36$ | $3.35 \pm 0.30$ |
| | 5% | $13.56 \pm 0.58$ | $10.00 \pm 1.10$ | $7.95 \pm 1.08$ |
| | 10% | $15.18 \pm 0.34$ | $13.05 \pm 0.81$ | $10.26 \pm 0.93$ |
| Skin (Bhatt & Dhall, 2012) | 1% | $0.04 \pm 0.01$ | $0.04 \pm 0.01$ | $0.05 \pm 0.03$ |
| | 5% | $0.15 \pm 0.03$ | $0.13 \pm 0.01$ | $0.10 \pm 0.02$ |
| | 10% | $0.15 \pm 0.03$ | $0.17 \pm 0.03$ | $0.13 \pm 0.02$ |

