# OpenReview forum: "Big Data, Leverage Scores, and Minimum Volume Covering Ellipsoids: Bridging Theory With Practice"
_ICLR.cc/2026/Conference — ICLR 2026 Conference Withdrawn Submission_

### Official Review · Reviewer_QZRi · 2025-10-27

**Soundness:** 3
**Presentation:** 3
**Contribution:** 4
**Rating:** 6
**Confidence:** 4

**Summary:**

This paper proposes a new approach to the computational bottleneck of solving MVCE in high-dimensional and large sample settings. The authors come up with an algorithm based on deterministic leverage score sampling. This algorithm is inspired by the idea of random numerical linear algebra and can be used to approximate the solution of MVCE efficiently. The work provides new theoretical results, including the upper bound for the initial and final optimal gaps, and shows that this method is significantly better in computational efficiency than existing approaches, such as Wolfe-Atwood and Fixed-Point algorithms.

**Strengths:**

Then main strength of this paper is that it proposed a novel and practical deterministic sampling framework, used to solve large-scale MVCE problem efficiently. The authors provide rigorous and comprehensive theoretical analysis. The results tell that this work does a better job than existing methods in terms of execution time and computational resources, maintaining near-optimal accuracy. It establishes a positive relationship between theory and practice, revealing authors’ deep understanding of the questions and rigorous academic attitude. Generally, this is great work on both theoretical quality and research value.

**Weaknesses:**

The main weakness lies in the experimental evaluation, which relies primarily on synthetic datasets. There is not enough validation on complex data from real world. In addition, the paper assumes that leverage scores follow a power-law decay, an assumption that may not hold in practice and is not thoroughly discussed in terms of robustness. Finally, the paper lacks straightforward geometric explanation, which could really make the paper more understandable.

**Questions:**

According to your theoretical analysis, the algorithm is efficient when leverage scores follow a power-law decay. However, experiments show that even though in a Gaussian dataset where leverage score approximately obeys uniform distribution, deterministic sampling is significantly better than uniform sampling, which is counter-intuitive. Could you please explain theoretically or intuitively why it is the best strategy to choose the point with the highest leverage score even in this situation?

---

### Official Review · Reviewer_xYTe · 2025-10-30

**Soundness:** 3
**Presentation:** 3
**Contribution:** 2
**Rating:** 4
**Confidence:** 4

**Summary:**

This paper tackles the high computational cost of the Minimum Volume Covering Ellipsoid (MVCE) problem in the "big data" regime where $n \gg d$. The standard complexity for many MVCE algorithms scales as $\mathcal{O}(nd^2)$. The authors propose a "sketch-and-solve" heuristic:

1.  Compute approximate statistical leverage scores for the $n$ data points.
2.  Select a small subset of $s$ points by sampling the $s$ rows with the highest scores.
3.  Solve the smaller $s \times d$ MVCE problem using an existing algorithm, e.g., Wolfe-Atwood algorithm.

The paper's main contributions are:
* Theoretical guarantees on the initial (Theorem 3) and final (Theorem 5) optimality gaps of the solution obtained from the sampled subset.
* A "new simplified proof" of a known subspace embedding theorem (Theorem 2) for this sampling scheme.
* A claim that, under a "power law decay" assumption for the leverage scores, the complexity is reduced to $\mathcal{O}(nd \log n + poly(d, 1/\epsilon, 1/\delta))$.
* Strong numerical experiments on synthetic and real-world data showing significant practical speedups.

**Strengths:**

1. **Well-Motivated Problem:** The MVCE problem is a classic in optimization, statistics, and computational geometry with wide-ranging applications. Scaling it to large datasets ($n \gg d$) is a practical and important challenge.

2. **Theoretical Guarantees:**  The paper provides the explicit bounds (Theorem 5) for the quality of an MVCE solution obtained via this specific sampling method. The final gap bound $g^* - g_s^* < d \log(\frac{1}{1-\epsilon})$ is clean and depends only on $d$ and the embedding quality $\epsilon$.

3. **Strong Empirical Results:** The numerical experiments are a clear strength. The proposed method vastly outperforms randomized leverage score sampling in terms of solution quality (optimality gap). The computation time is dramatically reduced.

**Weaknesses:**

1. **Limited Theoretical Novelty:** This paper lacks technical contribution.
    - The leverage score sampling is a known and very standard technique used in many theory papers.
    - The "sketch-and-solve" framework is also standard in the literature.
     - The "new simplified proof" of  a known subspace embedding theorem (Theorem 2) is a three-line argument based on a known matrix inequality ($x_i x_i^T \le l_i(X) X^T X$). This should not be stated as one major theoretical contribution.
    - The main theoretical results (Theorems 3 and 5) are essentially straightforward translations of the standard subspace embedding guarantee from Theorem 2 into the $\log \det$ objective function of the MVCE dual problem.

2. **Over-reliance on the "Power Law" Assumption.** The headline complexity claim of $\mathcal{O}(nd \log n + poly(d))$ is *entirely dependent* on the assumption that the leverage scores exhibit a power law decay. This is a very strong assumption on the input data.
    * The improved complexity relies on $s = poly(d, 1/\epsilon)$ (derived in Corollary 8).
    * If this assumption does not hold (e.g., for data with uniform leverage scores), the theory breaks down. Based on Algorithm 1, $s$ is the number of points required to capture $d-\epsilon$ of the total leverage score sum. For a uniform distribution, $l_i(X) \approx d/n$, which implies $s \approx (1-\epsilon/d)n$.
    * In this common case, $s = \mathcal{O}(n)$, and the complexity becomes $\mathcal{O}(nd \log n + nd^2(\log \log d + \delta^{-1}))$. This is *worse* than the $\mathcal{O}(nd^2)$ baseline the paper claims to improve upon. The paper's primary complexity claim is thus not a general result, but a special-case one.

3. **Theory fails to explain practice.** This is the paper's most significant flaw. The paper's title claims to ''bridge theory with practice'', yet the theory *cannot* explain the practice. The paper's numerical results on a Gaussian dataset show near zero optimality gap and a massive speedup (39 hours down to ~1000s). However, the paper explicitly states that the Gaussian dataset "has leverage scores that are close to uniform". As stated in W2, the paper's theory (which relies on power law decay) predicts that the method should *fail* on this dataset, requiring $s \approx n$ and offering no speedup. Please correct me if I'm wrong.

4. **Inadequate Comparison to State-of-the-Art.** The paper's main theoretical complexity comparison is in Section 4.2, where it argues for its superiority over the "Fixed Point algorithm" of Cohen et al. (2019). However, Woodruff & Yasuda (2024) also gives the algorithm with $\tilde{\mathcal{O}}(nd)$ time which out performs Cohen et al. (2019).

Woodruff, D. P., & Yasuda, T. (2024). John ellipsoids via lazy updates. NeurIPS 2024

**Questions:**

Please see the Weakness. If the authors could address my concerns, I will consider rasing the score.

---

### Official Review · Reviewer_bBt2 · 2025-10-30

**Soundness:** 3
**Presentation:** 2
**Contribution:** 2
**Rating:** 4
**Confidence:** 3

**Summary:**

The paper considers the Minimum Volume Covering Ellipsoid(MVCE) problem with observations in dimensions and is large. Assuming the leverage scores follow a power law decay, they show that the computational complexity of computing the approximation for MVCE is reduced from $O(nd^2)$ to $O(nd + poly(d))$. Their algorithm simply picks the rows with the highest leverage scores. Using the their assumption they show that the remaining rows have a very small contrivbution to the target function. They also give some numerical experiments to support their theoritecial claims.

**Strengths:**

All results are proven and there are also experiments supporting the claims.

**Weaknesses:**

Most of the paper is based on previous works.

The assumption that leverage scores follow a power law decayis not defined. Further this is a fairly strong assumption.

**Questions:**

no questions

---

### Official Review · Reviewer_ceB3 · 2025-10-31

**Soundness:** 3
**Presentation:** 3
**Contribution:** 3
**Rating:** 4
**Confidence:** 4

**Summary:**

The authors propose an approximation algorithm for computing the Minimum Volume Covering Ellipsoid (MVCE) of n points in d dimensions. The MVCE problem arises in a plethora of computational scenarios and it is an important computational paradigm. The proposed algorithm for MVCE is effective in the scenario where the number of points is much larger than the number of dimensions and satisfy an additional power law property. The proposed algorithm is based on a well-established concept in randomized linear algebra known as leverage scores. In particular, the leverage scores of the matrix containing the n points as rows is first approximated. Next, using the deterministic leverage score sampling algorithm of (Papailiopoulos et al., 2014), a small subset of s points is (deterministically) selected and on these s points the MVCE is calculated. The leverage scores are approximately computed using (Drineas et al. 2012) in the previous step. Selecting s points using a leverage scores based procedure provides a good approximation (coreset) for the MVCE problem. In addition, the authors provide experimental results on three datasets, the Rotated Cauchy, the Lognormal and a Gaussian dataset. The authors compare three selection strategies of the input points based on Algorithm 1, leverage score random sampling and uniform sampling. On these strategies, they experimental evaluate the optimality gap of the MVCE problem.

**Strengths:**

- The paper is well-written and the main contributions are clearly stated.
- While the leverage-score techniques used in the paper are well-established in the randomized numerical linear algebra literature, applying these techniques in the MVCE problem is interesting and of great value.
- Numerical experiments are provided (even on real data)
- The theoretical analysis is supported with rigorous proofs.

**Weaknesses:**

Based on my initial review of the paper, I have the following concerns.

1. Currently, in the abstract (L19), the running time of the algorithm does not explicitly depend on the approximation error. The authors should present the computational complexity for a given accuracy parameter $\epsilon > 0$.
2. It is not clear to me how does the power law decay affect the number of rows selected in Algorithm 1, and how restrictive this assumption is.
3. It is highly unclear how the problem is connected to learning representations, or machine learning in general. The problem seems interesting but there is no attempt to tie it to the topic of the conference. (This specific concern did not affect my score, but I think it is quite important).
4. The experimental evaluation could be performed in larger datasets - the paper title mentions Big Data. The datasets used in the experiments have 10^9 entries ~ 4-8GBs, which is not tiny, but it can easily fit in a personal laptop.
5. There are a few other minor concerns, which can be probably easily solved, I list them below in "Additional feedback" under "Main Questions".

**Questions:**

### Main Questions
The following questions would help me to understand the paper better and give a provide a more informed final recommendation.
1. How does the approximation error affect the running time of the algorithm? approximation-complexity trade-offs.
2. How does the power law decay affect the number of rows selected in Algorithm 1? In particular, when does your method become inferior compared to Cohen et al. 2019?
3. How restrictive the power law assumption on the leverage scores is? Are there any applications of MVCE where power law is typical? Or is the power law assumption an artifact of Algorithm 2 (Line 2).
4. How can you set $\epsilon$ on Algorithm 1? This parameter is quite important for the number of points selected, since you approximate the leverage scores. Can we assume that it is always a constant?
5. Section 4.2 (L246-253). This is also related to my first point (accuracy parameter). Shouldn’t $\delta$ by the accuracy parameter here? Why do you assume $\delta < 1/d$ in L246 ?
6. Bonus: why is MVCE important for machine learning?


### Additional Feedback
The following (mostly minor) comments could be helpful to further improve the manuscript.
- L47: “Deterministic Sampling” could be rephrased here. I understand that you inherit the term from Algorithm 1 but could you rephrase as selection? There is no sampling involved here.
- L48-L49: you may want to give a reference to leverage scores. Some readers might not be familiar with the concept.
- L53 (… a new simplified proof of the subspace embedding.. ). This is great but please provide a few more sentences about which parts you simplified. Is Lemma 4 in Cohen et al. 2015 used in McCurdy et al. 2019?
- Fig 1(a) and 2(a) the blue line misses the legend text.
- L225: \beta parameter here should be related to the accuracy parameter \epsilon
- L246-L253: Explicitly mention “Algorithm 2 from Cohen et al. 2019”. It might bring confusion with you Algorithm 2.
- Conclusion: “the first theoretical guarantees… on the quality of initial and final solution …” this is for the case of leverage score selection correct?
- In experiments, why did you use only exact leverage score values? How do the experiments look like with approximate?
- Why do you use a different machine for Appendix E ?

---

### Official Review · Reviewer_DYSx · 2025-11-01

**Soundness:** 3
**Presentation:** 3
**Contribution:** 2
**Rating:** 2
**Confidence:** 3

**Summary:**

The document discusses a new algorithm for solving the Minimum Volume Covering Ellipsoid (MVCE) problem using leverage score sampling. The main contribution is a theoretical understanding of the optimality gap when using sampling based on deterministic leverage scores. When the leverage scores decay sufficiently fast the computational complexity improves from $O(nd^2)$ to $O(nd+ poly(d))$ under the new setup. This is a big improvement in the sampling regime $n\gg d$ considered in this paper.

**Strengths:**

1. The main theoretical result on the optimality gap of initial and final solutions for the MVCE problem is very interesting, and the proof is presented in a relatively simple way.
2. Under the given assumptions, the improvement in the computational cost, from $O(nd^2)$ to $O(nd log(n))$ is a significant development.

**Weaknesses:**

1. The cost reduction depends on power law decay of the leverage scores. It is not clear when this decay law happens.
2. There are words such as initial and final optimality gap which to my understanding allude to some iterative method. It might be better to include the Wolfe-Atwood (WA) algorithm since someone like me is not familiar with it.
3. Novelty of this work is not clear. Is only the proof new?

**Questions:**

Questions are based on addressing the weaknesses above.
1. Can the authors describe assumptions behind the power law decay?
2. Is the Wolfe Atwood algorithm essential for these results? It might be better to show steps of this algorithm somewhere, without which "initial" and "final" seem to lack some context.
3. Can the authors describe if only the proof is the new contribution of this paper? Did such a proof exist earlier but was more complex?
4. The simulations involve a random sampling method. Can the authors describe why random sampling instead of deterministic is ever used in practice? In other words is there a trade-off to using deterministic sampling?

---

### Note · Authors · 2025-12-02

I have read and agree with the venue's withdrawal policy on behalf of myself and my co-authors.